# Effect of Long Working Hours on Depression and Mental Well-Being among Employees in Shanghai: The Role of Having Leisure Hobbies

**DOI:** 10.3390/ijerph16244980

**Published:** 2019-12-07

**Authors:** Zan Li, Junming Dai, Ning Wu, Yingnan Jia, Junling Gao, Hua Fu

**Affiliations:** Health Communication Institute, School of Public Health, Fudan University, Shanghai 200032, China; zanli17@fudan.edu.cn (Z.L.); 13524430617@163.com (N.W.); jyn@fudan.edu.cn (Y.J.); jlgao@fudan.edu.cn (J.G.); hfu@fudan.edu.cn (H.F.)

**Keywords:** long working hours, having hobbies, depression, mental well-being

## Abstract

Our aim is to examine the associations between long working hours and depression and mental well-being among the working population in Shanghai, as well as to identify the impact of having hobbies on these relationships. A cross-sectional study was conducted in Shanghai, with depression assessed by the Patient Health Questionnaire-9 (PHQ-9) scale and mental well-being assessed by the World Health Organization five-item Well-Being Index (WHO-5) scale. The phenomenon of long working hours (69.3%) was quite common among employees in Shanghai, and the rate of working over 60 h was 19.3%. Those who worked over 60 h had the highest prevalence of poorer mental health compared with individuals working ≤40 h per week. After adjustment in the logistic regression model, those who reported weekly working time over 60 h were 1.40 (95%CI: 1.03–1.90) and 1.66 (95%CI: 1.26–2.18) times more likely to have depression and poor mental well-being (PMWB), respectively. Adjusted ORs for having hobbies were 0.78 (95%CI: 0.62–0.97) and 0.62 (95%CI: 0.51–0.75), respectively. Meanwhile, having hobbies could significantly lower the mean score on the PHQ-9 and elevate the mean score on the WHO-5 in each working time group, with no interaction effect. Long working hours could have a significantly negative impact on workers’ psychological health. Importantly, having hobbies in their daily lives might help to mitigate the adverse effects of long working hours on workers’ depression and mental well-being.

## 1. Introduction

Individual overall health during working life can be determined by working conditions, and long working hours have caught the eye of public health professionals and labor policymakers. According to recent statistics from the Organization for Economic Cooperation and Development (OECD), the countries with the highest annual working hours per worker were Mexico, Costa Rica, and South Korea [1]. However, statistics about working hours in mainland China are limited. Nevertheless, the Chinese phenomenon called “guolaosi” (death by overwork) has happened in several enterprises and received vast media attention. As Chinese are educated to complete tasks with absolute dedication based on Confucian belief since their childhood, domestic workers are prone to overburdening or burning themselves out during career life. One earlier study using data from the China Health and Nutrition Survey (CHNS) from 1991–2009 found that the average weekly working hours of Chinese employees were around 47 h, with approximately 62% of respondents working more than the national standard of 40 h per week [2].

Considering the severity of long working hours’ prevalence, empirical studies have shown that long working hours can elevate the risk of several physical diseases, such as coronary heart disease, stroke, and diabetes [3,4,5]. Moreover, poor mental health among employees, such as depressive symptoms and suicidal thoughts, has also brought about tremendous disease burden in developed countries, leading to work disability and reduced quality of life [6,7,8].

Clarifying the relationship between long working hours and psychological outcomes has gained significant research interest, and there is a growing body of evidence suggesting that long working hours are detrimental to workers’ mental health and psychological well-being. Studies have shown that long working hours are implicated in anxiety, depression, suicidal thoughts, and worsened emotional well-being [9,10,11,12,13,14,15,16]. A prospective cohort designed in Australia over the period of 2001–2012, using the Mental Component Summary of the Short Form 36 (SF-36) survey to measure outcome, found that respondents had worse mental health when they were working 49–59 h or 60 h or more compared to those who were working 35–40 h/week [10]. A longitudinal study based on the Korea Welfare Panel Study (KOWEPS) conducted from 2010–2013, using the Center for Epidemiological Studies Depression (CES-D) scale to measure depressive symptoms, found that employees working over 68 h had higher odds of depressive symptoms after full adjustments compared with individuals working 35–40 h/week [12].

However, some other studies showed that the adverse effect of long working hours on mental health only occurred among either male or female participants [8,17,18]. Beyond this, the results of a nationwide prospective follow-up study conducted among Danish senior medical consultants did not support the hypothesis that long working hours may increase the risk of depression [19]. Furthermore, a systematic review and meta-analysis with an inclusion standard of prospective or cohort study design, confining the outcome as depressive disorders clinically diagnosed or assessed by a structured interview, showed that overtime work was associated with a small, non-significant, elevated risk of depressive disorder and concluded that the effect of long working hours on depressive disorder remained inconclusive [20].

Inconsistencies between past findings might be attributed to several factors, including the disparity in study design, different assessment tools of exposure, outcome, and potential confounding factors, various kinds of participant occupations, heterogeneity in sample size, insufficient controlling for bias, and even cultural discrepancy among regions. For example, the cut-off point chosen for long working hours had differential values, ranging from 40 h/week to as high as 70 h/week, and mental health status was measured with SF-36, CES-D, and WHO-5 scales.

Although long working hours may have adverse effects on the mental health status of the working population, positive life events could help employees cope better with the impact of adverse events. Noticeably, growing evidence has shown that leisure activities (e.g., hobbies, cultural activities, exercise, and sports) have salutary effects for individuals. Scholars in the leisure science field have long insisted that leisure plays a crucial role in promoting overall well-being and buffering stress [21]. These benefits may be consequences of leisure encouraging positive feelings and augmenting ways of utilizing social and physical resources, which ultimately help individuals to feel refreshed and better deal with various stressors [22,23]. For example, an online study in Australia reported that satisfying leisure activities were beneficial to dementia caregivers aged 18–82 years, and particularly engaging in social activities buffered the negative impact of caregiving [24].

In sum, research on the associations between long working hours and mental health status has remained very limited in China, and no research has demonstrated whether having hobbies could play a beneficial role in this relationship. Hence, the present study aimed to firstly explore the distribution of working hours among employees and examine how long working hours affect their depression and mental well-being using representative data from a large cross-sectional survey conducted in Shanghai, and secondly investigate the impact of having hobbies on this relationship.

## 2. Materials and Methods

### 2.1. Sampling and Participants

This survey was conducted using a cross-sectional methodology between July and August 2018. Shanghai is one of the boom cities in eastern China, and employees live a fast-paced life with a great of hustle and bustle. Excessive dedication to work could make them vulnerable to mental health problems due to a work–life imbalance and lack of recovery after busy routines.

To make up as representative a sample as possible, we used a multistage random sampling scheme containing a variety of work types in Shanghai, including white-collar workers, blue-collar workers, service personnel, and self-employed industrialists. In the first step, our sampling was done by geographical location, covering one downtown area of Xuhui District, and the other five districts (Pudong, Minhang, Songjiang, Qingpu, and Jiading Districts) which are located around the urban district. In the second step, sampling was done by the workplace or organization, including participants from seven companies or workplaces (staff number ≥300) based on our previous recruitment. Lastly, in the third step, sampling was done by the unit or department, taking employees from a specific work unit within the company or workplace as our study subjects. For this study, an initial number of 3310 employees were recruited. We designed a questionnaire to collect demographic and work characteristics data and used the PHQ-9 scale and WHO-5 scale to assess depression and mental well-being, respectively. Previously designed questionnaires were distributed to various departments by investigators and company administration managers contacted. Trained investigators were allocated to instruct participants when they had any questions and to ensure good quality control during our survey. All the participants completed the questionnaires independently with informed consent. Finally, 2985 valid questionnaires were collected with a response rate of 90.2%.

### 2.2. Exposure and Other Variables

The exposure variable was the average number of working hours employees worked per week. Participants answered the question “How many hours do you currently work on average per week?” Long working hours (defined as overtime per week beyond normal or regular working hours) was operationally defined as more than 40 h/week in this study, which was in line with the definition adopted by many countries, according to a recent International Labor Office survey and previous studies [2,12,19,25]. Then weekly working hours (WWH) were split into four groups (≤40 h, 40 h < WWH ≤ 50 h, 50 h < WWH ≤ 60 h, and >60 h), with ≤40 h used as the reference category.

We also collected other information for our analysis. The questionnaire included basic sociodemographic characteristics, questions about living and working conditions, self-perceived health (SPH), and health-related behaviors. For having or not having hobbies, participants answered the question “Do you have any regular hobbies or leisure activities?” For SPH, respondents responded to the question “In general, how would you rate your overall health?” on a scale ranging from 1 to 5, indicating “very poor,” “poor,” “fair,” “good,” and “very good,” respectively. Work-related psychosocial factors (job demand, job control, and social support) were obtained with the Job Content Questionnaire based on Karasek’s job demand-control model, and job stress was calculated via job demand divided by job control. Moreover, gender, age, education level, household registration (HR), marital status, monthly income, living with family, sleeping status, drinking, smoking status, and moderate physical activity (MPA) per week were addressed as categorized variables.

### 2.3. Outcome Variables

Depression and mental well-being were measured using the PHQ-9 (Cronbach’s alpha = 0.869) and WHO-5 (Cronbach’s alpha = 0.926), respectively. The PHQ-9 scale has been proved valid and reliable in Chinese populations [26]; in this scale, participants are asked to rate the frequency of specific depression symptoms over the last two weeks from not at all (score 0) to nearly every day (score 3), for a total score ranging from 0 to 27. At a cutoff score of 9 (PHQ-9 score ≥ 10), the PHQ-9 scale was found to have a sensitivity of 88% and a specificity of 88% for detecting major depression compared with a structured psychiatric interview [27]. The WHO-5 scale has also been shown to be a well-being scale with good reliability and validity, serving as a sensitive screening test for mental well-being, and has been used in Chinese populations [28]. Participants were asked to rate their status from never (score 0) to all the time (score 5) over the last four weeks, and the sum score could range from 0–25, with a total score below 13 defined as poor mental well-being (PMWB).

### 2.4. Statistics

The WHO-5 scores were calculated according to the established scoring algorithms, with higher scores indicating better mental well-being, contrary to the depression scores of the PHQ-9. Sociodemographic variables were categorized for analysis purposes as shown in Table 1. We first investigated the prevalence of depression and PMWB across different characteristics among participants by using Pearson’s *χ*^2^ test. Then the associations between long working hours and psychological outcomes (depression and PMWB) were assessed via multivariate logistic regression. The logistic regression models were fitted using the categories of scale scores as the dependent variables and weekly working hours as the independent variable. The odds ratios (ORs) and 95% confidence intervals (CIs) were calculated. Finally, we assessed the impact of having hobbies on the mean score changes of the PHQ-9 and WHO-5 scales in the four WWH groups using a general linear model and simultaneously explored whether there was an interaction effect between working hours and having hobbies. A two-tailed alpha with *p* value < 0.05 was considered statistically significant. All the data analyses were performed using SPSS software for Windows, version 22.0 (IBM, Armonk, NY, USA).

### 2.5. Ethics Approval

Informed consent was obtained from the participants after detailed explanation of this study. The study was approved by the Institutional Review Board (IRB) of the School of Public Health of Fudan University with an IRB approval number of IRB #2015-12-0574.

## 3. Results

### 3.1. Basic Findings

In this study, a valid sample size of 2985 was confirmed, and there were more female participants (54.6%) than male participants (45.4%). The mean age was 30.98 years (SD = 9.48), with a range of 18–60. Among all the employees, the prevalence of long working hours was 69.3%. In detail, the proportions of those who worked 41–50 h, 51–60 h, and more than 60 h per week were 20.6%, 29.4%, and 19.3%, respectively. The mean scores on the PHQ-9 and WHO-5 were 6.09 (SD = 4.47) and 16.09 (SD = 5.66), respectively. According to our cut-offs, the prevalence of depression and PMWB was 19.0% and 25.3%, respectively.

The associations between general characteristics and depression and PMWB are shown in Table 1. As expected, workers who had hobbies had significantly better mental health status than those who did not in terms of both prevalence of depression (16.4% vs. 21.6%) and PMWB (20.4% vs. 30.7%). In terms of weekly working hours, the prevalence of PMWB gradually increased with workers working from ≤40 h to >60 h per week, but this was not the case for depression except that working over 60 h per week significantly predisposed workers to the highest prevalence of depression (25.9%). As for work-related psychosocial factors, workers who had low job control, low social support, and job stress had a significantly higher prevalence of depression and PMWB.

### 3.2. Factors Related to Depression and PMWB

Table 2 represents the multivariate logistic regression results. Under the crude model (Model I), there were significant associations between long working hours, having hobbies, and poor mental health status, with increased ORs of 1.62 (95%CI: 1.26–2.09) and 1.81 (95%CI: 1.44–2.29) for working over 60 h per week on depression and PMWB, respectively. Similarly, decreased ORs of 0.72 (95%CI: 0.59–0.86) and 0.58 (95%CI: 0.49–0.68) were found for having hobbies. However, when the adjusted model (Model II) was run after adjustment for basic characteristics, health-related behaviors, and work-related psychosocial factors, the ORs for working over 60 h per week declined to 1.40 (95%CI: 1.03–1.90) and 1.66 (95%CI: 1.26–2.18), respectively, and the ORs for having hobbies climbed to 0.78 (95%CI: 0.62–0.97) and 0.62 (95%CI: 0.51–0.75).

### 3.3. The Role of Having Hobbies

Table 3 shows the mean scores of two measurements stratified by having hobbies or not in the general linear model. In all four WWH groups, the mean scores on the PHQ-9 for having hobbies were significantly lower than those for having no hobbies. Accordingly, the mean score on the WHO-5 in each WWH group was significantly higher among workers who had hobbies. Moreover, the mean score on the WHO-5 decreased with increasing weekly working hours for both those who had hobbies and those who did not. However, this was not the case for the PHQ-9, for which the mean scores of the 41–50 h group were lower than those of the ≤40 h group. P values for the interaction effects between weekly working hours and having hobbies on the PHQ-9 and WHO-5 demonstrated statistically insignificant results (Figure 1).

## 4. Discussion

The main findings of our study were that long working hours, specifically working over 60 h per week, were significantly associated with depression and PMWB among the working population in Shanghai and having hobbies in their lives played a beneficial role in these relationships without an interaction effect.

In this study, the prevalence of long working hours among participants was 69.3%, which was slightly higher than that (62.0%) of one previous study using Chinese national data from 1991 and 2009 [2], indicating that employees in Shanghai are more prone to working overtime due to the fast-pace life of this megacity. We also found that nearly one-fifth of respondents had depression, and approximately one-fourth showed PMWB. We found that working over 60 h in one week was an independent risk factor for both depression and PMWB, which was similar to one study using the WHO-5 in Korea [14] and others using different measurements [11,12,16]. However, we did not detect a dose-response relationship between weekly working hours and risks of depression or PMWB, as some previous studies did [13,16].

Long working hours can affect the overall health of workers from different perspectives. Regarding the relationship between long working hours and mental health status, there could be complicated pathways through which the effect of long working hours may give rise to deterioration in psychological health or mental well-being. Furthermore, long working hours may partially exert an impact on mental health via measured or unmeasured factors in our study. Some studies with robust evidence have shown that the existence of long working hours could lead to a shortage of sleep and time to “recover or repair” from the demands of a job, making workers more vulnerable to worsening mental health [29,30]. In addition to the loss of time for sleep, long working hours may cut down the time available for other leisure-related activities or personal hobbies that help to relieve pressure and provide refreshment, further causing a decline in mental health [9]. It is also possible that workers with long working hours cannot maintain a balance between work and family life, often leading to time poverty and exclusion from family events. Consequently, work–family conflicts come into being, which in turn results in higher stress and depression [31]. Previous studies have revealed that longer working hours are associated with work–family conflict, which contributes to less family interaction, more conflict in marriage, and less participation in housework [32]. Besides, long working hours could probably result in worse health behaviors and have been associated with increased risk of alcohol use and suicide [11,33]. Noticeably, leisure hobbies are indeed about employees’ private lives. When employers or managers are considering about conducting any workplace wellness (or well-being) programs, they should take care to avoid the tendency to shift blame from institutions/employers to individuals/employees, to create and sustain hierarchy, and to cede employers control over employees’ private lives [34].

The present study is one of the first to clarify how long working hours affect mental health status among the occupational population in China and the first to provide evidence of the beneficial role of having hobbies in this relationship. This study also provides a better understanding of the current situation of long working hours and the prevalence of poor mental health status in Shanghai, China. However, there are also several limitations to this study. First, it was a cross-sectional study, unable to reliably establish a temporal relationship between long working hours and mental health outcomes. A longitudinal study would be necessary to verify whether long-term exposure to long working hours could have effects on the causal change of mental health status or psychological well-being. Second, a lack of validity from self-reported surveys may exist. The use of self-reports to assess and categorize the number of working hours may give rise to subjective errors. Third, when touching on sensitive issues, such as depression in a domestic context, respondents tend to answer questions more in line with social norms than with their actual situations—A phenomenon known as social desirability [35], in which depressive individuals may be reluctant to admit their psychological problems even under conditions of anonymity and confidentiality. In addition, there may have been a healthy survivor effect, as individuals with severe depression or the worst state of psychological well-being are considerably unlikely to be present at work and have certain difficulties in maintaining full-time occupations. Finally, this study was conducted in Shanghai, a coastal megacity where employees may be more likely to face a fast-paced work life and enormous pressure to get promoted, and we excluded some job categories, such as part-time jobs and shift work. Hence, the current findings may not be highly generalizable to the whole working population in China.

## 5. Conclusions

Using two types of outcome measurements, we found that long working hours (over 60 h/week) were uniformly associated with a decline in mental health status compared to standard working hours. Having hobbies played a beneficial role in this relationship for working-age people. These findings have important implications for workplace policy and the prevention of mental illness among workers. Strengthening regulation of long working hours and improving time management skills among working-age people in workplaces, while encouraging employees to cultivate certain kinds of hobbies or leisure activities in their spare time, may be conducive to their mental health status. Workplace health promotion programs or interventions should also pay close attention to these factors to achieve the goal of total worker health. Future studies should adopt a longitudinal methodology to identify the causal relationships between long working hours and psychological outcomes, include a more representative sample with higher generalizability, and take examine hobbies in more detail to determine which one can best enhance workers’ mental health.

## Figures and Tables

**Figure 1 ijerph-16-04980-f001:**
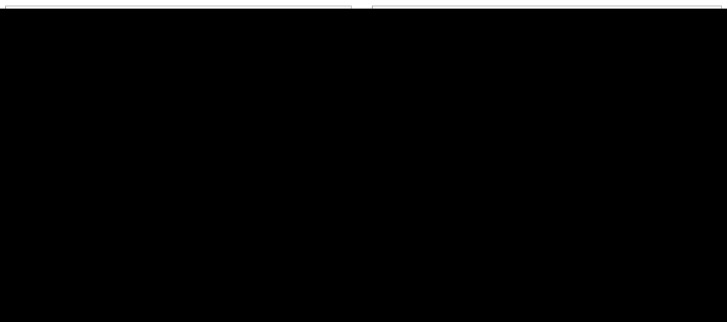
(**a**) Mean score trend of PHQ-9 in four WWH groups by having hobbies or not; (**b**) Mean score trend of WHO-5 in four WWH groups by having hobbies or not.

**Table 1 ijerph-16-04980-t001:** Prevalence of depression and poor mental well-being (PMWB) by characteristics of participants.

Variables	Depression *n* (%)	PMWB *n* (%)	*χ* ^2^	df	*p*
Gender			0.007; 2.92	1	0.485; 0.048
Male	248 (19.2%)	316 (24.1%)			
Female	298 (19.1%)	423 (26.9%)			
Age			3.61; 3.00	3	0.307; 0.392
15–24	161 (19.5%)	220 (26.4%)			
25–34	245 (19.1%)	333 (25.6%)			
35–44	91 (20.5%)	115 (25.6%)			
≥45	58 (15.6%)	81 (21.8%)			
Education level			1.15; 5.27	3	0.765; 0.153
Junior high school	138 (19.0%)	199 (27.4%)			
Senior high school	159 (19.3%)	194 (23.3%)			
College	107 (17.4%)	152 (24.2%)			
Bachelor or above	139 (19.5%)	197 (27.3%)			
HR			0.72; 8.85	1	0.215; 0.003
Local	105 (17.8%)	122 (20.6%)			
Migrant	449 (19.3%)	627 (26.6%)			
Marital status			22.14; 1.95	2	<0.001; 0.378
Unmarried	253 (20.5%)	329 (26.3%)			
Married	276 (17.4%)	395 (24.7%)			
Divorced and others	19 (44.2%)	14 (31.8%)			
Monthly income (¥)			2.93; 2.55	3	0.402; 0.466
<3000	101 (21.5%)	132 (27.9%)			
3000–6000	305 (18.4%)	408 (24.4%)			
6001–10,000	103 (17.6%)	154 (25.9%)			
≥10,000	36 (18.8%)	48 (25.0%)			
Living with family			3.51; 3.10	1	0.060; 0.079
Yes	194 (17.1%)	277 (24.3%)			
No	321 (20.0%)	445 (27.2%)			
SPH			62.06; 190.43	1	<0.001; <0.001
Good	155 (12.5%)	159 (12.7%)			
Bad	392 (24.2%)	580 (35.2%)			
Sleeping status			22.41; 71.17	1	<0.001; <0.001
Good	218 (15.3%)	263 (18.4%)			
Bad	328 (22.2%)	477 (31.9%)			
Drinking			13.26; 3.06	1	0.001<; 0.084
Yes	168 (23.5%)	201 (27.6%)			
No	376 (17.4%)	533 (24.4%)			
Smoking			16.90; 10.04	1	<0.001; 0.002
Yes	131 (25.2%)	161 (30.8%)			
No	413 (17.4%)	578 (24.1%)			
MPA/week			2.15; 0.13	1	0.150; 0.721
Yes	370 (19.9%)	470 (25.0%)			
No	179 (17.6%)	262 (25.6%)			
Having hobbies			12.50; 41.08	1	<0.001; <0.001
Yes	241 (16.4%)	304 (20.4%)			
No	310 (21.6%)	443 (30.7%)			
WWH			23.44; 28.82	3	<0.001; <0.001
≤40 h	158 (17.7%)	200 (22.1%)			
40 h < WH ≤ 50 h	96 (15.8%)	144 (23.6%)			
50 h < WH ≤ 60 h	153 (17.9%)	211 (24.4%)			
>60 h	148 (25.9%)	194 (33.9%)			
Job demand			6.36; 3.04	1	0.012; 0.088
Low	229 (17.0%)	327 (24.0%)			
High	318 (20.7%)	414 (26.8%)			
Job control			24.18; 45.70	1	<0.001; <0.001
Low	343 (22.4%)	472 (30.6%)			
High	206 (15.2%)	270 (19.7%)			
Social support			124.46; 109.92	1	<0.001; <0.001
Low	391 (27.3%)	491 (33.9%)			
High	162 (11.0%)	254 (17.1%)			
Job stress			39.27; 49.26	1	<0.001; <0.001
No	254 (15.7%)	363 (22.1%)			
Yes	202 (26.5%)	273 (35.7%)			

**Table 2 ijerph-16-04980-t002:** The odds ratios and 95% confidence intervals of the relationships of weekly working hours with depression and PMWB.

	Factors	ORs of Depression	ORs of PMWB
Model I	WWH		
	≤40	Reference	Reference
	41–50	0.87 (0.66–1.15)	1.09 (0.86–1.39)
	51–60	1.01 (0.79–1.29)	1.14 (0.91–1.42)
	>60	1.62 ** (1.26–2.09)	1.81 * (1.44–2.29)
	Having hobbies		
	No	Reference	Reference
	Yes	0.72 ** (0.59–0.86)	0.58 ** (0.49–0.68)
Model II	WWH		
	≤40	Reference	Reference
	41–50	0.91 (0.67–1.25)	1.14 (0.86–1.51)
	51–60	0.93 (0.69–1.25)	0.95 (0.76–1.28)
	>60	1.40 * (1.03–1.90)	1.66 ** (1.26–2.18)
	Having hobbies		
	No	Reference	Reference
	Yes	0.78 * (0.62–0.97)	0.62 ** (0.51–0.75)

Note: Model I: Crude logistic model only including weekly working hours (WWH) or having hobbies; Model II: Adjusted logistic model including full factors, adjustment for demographics, health behaviors, and working conditions; * Statistically significant at *p* < 0.05.; ** Statistically significant at *p* < 0.0001.

**Table 3 ijerph-16-04980-t003:** Mean scores on PHQ-9 and WHO-5 in four WWH groups by having hobbies or not.

WWH	Mean Score on PHQ-9		Mean Score on WHO-5		*F* (*F1*; *F2*)	*p* (*p1*; *p2*)
	Having Hobbies	No Hobbies	Having Hobbies	No Hobbies		
≤40 h	5.67 ± 4.49 (*n* = 476)	6.30 ± 4.70 (*n* = 403)	17.60 ± 5.38 (*n* = 484)	16.07 ± 5.70 (*n* = 412)	4.14; 17.16	0.042; <0.001
41–50 h	5.22 ± 3.97 (*n* = 316)	6.11 ± 4.22 (*n* = 291)	16.86 ± 4.99 (*n* = 318)	15.33 ± 5.77 (*n* = 291)	7.04; 12.34	0.008; <0.001
51–60 h	5.71 ± 4.19 (*n* = 418)	6.29 ± 4.30 (*n* = 430)	16.50 ± 5.04 (*n* = 430)	15.24 ± 5.70 (*n* = 430)	4.02; 11.71	0.045; 0.001
>60 h	6.32 ± 4.74 (*n* = 256)	7.38 ± 4.95 (*n* = 312)	15.69 ± 5.98 (*n* = 259)	14.32 ± 6.19 (*n* = 310)	6.76; 7.23	0.010; 0.007

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
