# Peer review of "Effect of Long Working Hours on Depression and Mental Well-Being among Employees in Shanghai: The Role of Having Leisure Hobbies"

_ijerph, 2019, doi:10.3390/ijerph16244980_

Round 1

Reviewer 1 Report

Overall comments

The study investigates a topic that could be of interest for the scientific community. However, several improvements are suggested. For more details, please see below.

Specific comments

Background

After had read the background it is not still clear why the study was important to conduct and what gap in the scientific literature it would contribute to. A meta-analysis is referred to (no 20 in the reference list) and several of the arguments that are brought up there could be useful for this study to build upon.

The background would benefit from a clear and logic thread used from start to the aim to build up the argumentation for the study. It is for example not clear why it is important to narrow down to China already in the beginning of the background description. Is this topic not relevant for other countries?

The argumentation for why leisure activities could be relevant as a mediating factor needs to be strengthened. It is for example no transition from the end in the section about work hours and depression to the section about leisure activities.

The first sentence starts with “The introduction should…”. This text needs to be removed.

The aim, as described in the background, differs from that described in the abstract. Please check so you are consistent. For example, in the abstract you write that you will examine the associations between long working hours and depression /…/, whereas in the background section you write that you will examine how long working hours affect their mental health status.

Materials and methods

Please clarify the data collection procedure. It is difficult to follow the different steps in the multistage random sampling scheme. Was the first questionnaire used to make sure that the included sample contained a variety of workplaces or groups of employees? That the sample included for example employees with mental ill-health or long working hours? A multistage random sampling procedure often means that you start with a larger sample and takes smaller and smaller sample at each stage. If that was the procedure used, an easy step by step presentation would be good to have.

Are the questionnaires used validated? For example, the questions related to working hours, working conditions, leisure activities and health? If yes, please add this information.

Please add references to the different questionnaires used, the method being referred to and so on.

From the formulation of the aim it seems like leisure activities is considered as a mediating variable. However, the analyses are not conducted as recommended if mediation is tested for.

Working conditions has previously mentioned as variables to control for in the analyses. In table 2 the results of the logistic regression are presented including the results for different factors related to work conditions. As this is not part of the aim, this does not need to be presented in the table. A footnote including information about that it was controlled could be used.

Is it possible to add information about number of employees in each group in table 3? It would be interesting to know if the result depends on small groups.

Discussion

This section could be cleaned from texts related to other parts than described in the aim, for example row 217-222.

Author Response

Dear Reviewer:

Thank you for your valuable and helpful comments and suggestions. We have studied comments carefully and made corrections which we hope meet with approval. Revised portions are marked in the revised paper. Please see the attached for more details.

Best regards,

Zan Li

Reviewer 2 Report

This was a very interesting article. It provides a good quality background of information and establishes a sound justification for the study. The Methods are well described and the results are displayed in a very understandable fashion. There are just a few edits that I would make and one question about the sample that I would like to see answered within the context of this article. 

First, the first sentence in the introduction is not clear or understandable. If you take it out, the rest of the introduction makes much more sense to the reader. 

There are some very big paragraphs in the literature review so I suggest starting at least two additional paragraphs during this section. Beginning on line 46, beginning with the word, "Clarifying", it may help to start a new paragraph at this point. 

Again, on line 58, beginning with the word, "However" it would also be helpful to see another paragraph established.

Going down to the way you describe your sample, can you please describe how your sample of women at 54.6% is reflective of how many women are in the workforce in the population? How close to the true population parameters is 54.6% female populace? That is, are there really more women than men in the workforce in the area under investigation? If not, why would the sample not be weighted to reflect true population parameters? 

Also, I believe that further discussion should be made for investigating which types of hobbies may have the most impact on mental health, depression, overall well-being. You have used a very broad category, "whether or not they have established hobbies or leisure" but the reader knows nothing about what types of hobbies are even typically enjoyed in this population area. Please either give a description of what we might expect as general hobbies undertaken in this area or provide a recommendation for assessing specific hobbies for their impact on mental well being in future studies. I see this as one of the most striking limitations to the study in that it does not specify hobby / leisure type and we do not know if these are more artistic in nature (such as painting) or physical in nature (such as jogging, exercise) related and so the pathways between leisure / hobbies to well-being are not very clear and need to be further investigated. 

Author Response

(The authors gave the same response as above.)

Reviewer 3 Report

The problem I and many other commentators have about workplace wellness (aka well-being) research, programs, initiatives, and ideology in general is the tendency to shift blame from institutions/employers to individuals/employees; to create and sustain hierarchy; cede employers control over employees’ private lives; and institutionalize disability bias.

I appreciate that the authors have largely problematized employers by critiquing long work hours. Their focus on hobbies, however, problematizes individual employees, and may have an unintended consequence of ceding employers control over employees’ private lives.

The results suggest that longer WWH, low job control, and job stress might have something to do with (perhaps have a causal relationship) with poor well-being. Yet the authors, while problematizing the longer WWH (job control and job stress were not their focus), also problematize what employees do in their spare time—their private lives.

Accordingly, I have concerns about the following statements:

“having hobbies in their daily lives might help to mitigate the adverse effects of long working hours on workers’ depression and mental well-being.” “examine hobbies in more detail to determine which one can best enhance workers’ mental health.” “improving time management skills among working-age people in workplaces, while encouraging employees to cultivate certain kinds of hobbies or leisure activities in their spare time, may be conducive to their mental health status.”

Maybe these relationships might hold up with more research, maybe not. How much, if any, causal relationships exist, who knows? But I have concerns about encouraging employers to be involved in employees’ private lives.

The statement about time management skills is problematic for some of the same reasons, but also because they may stigmatize workers with specific learning disorders (learning disabilities), attention-deficit/hyperactivity disorder (ADHD), alternative working styles, in addition to workers who are simply very dedicated to their profession. Also note that encouraging employers to set caps on workers’ hours can always result in employers coercing employees not to report working long hours or making them work in all but name at home or unofficially. Encouraging them to reduce work load, however, might not result in these potential unintended adverse consequences.

I do think, however, that the data and results on longer WWH, low job control, and job stress are important, and that the results for hobbies are also important in one particular respect. It has often been suggested that specific lifestyle practices and activities popular among cultural elites (e.g., exercise, kale consumption, yoga) are effective at maintaining well-being and preventing depression. (The suggestions are often made by researchers who are themselves cultural elites with time to exercise, practice yoga, and eat kale.) Yet it is also quite possible that whatever mental health benefits derive from engaging in activities (whatever they may be) that confer on people a sense of accomplishment and agency. This is suggested by research, for example, on supported employment for persons with severe mental illness.

Should the authors revise, they should take care to avoid the problems I outlined in the first paragraph of my comments; stress that employers should not get involved in the private lives or monitor their employees; and refocus their discussions of hobbies.

They should see also:

Kirkland A. Critical perspectives on wellness. Journal of health politics, policy and law. 2014 Oct 1;39(5):971-88.

Author Response

(The authors gave the same response as above.)

Round 2

Reviewer 2 Report

I have read over the authors responses to my original feedback and am satisfied they have made the edits needed for this to be published. 

I am also satisfied with English corrections that have already been made.